# Constructing a Low–Cost Si–NSs@C/NG Composite by a Ball Milling–Catalytic Pyrolysis Method for Lithium Storage

**DOI:** 10.3390/molecules28083458

**Published:** 2023-04-14

**Authors:** Qi Zhang, Ning-Jing Song, Can-Liang Ma, Yun Zhao, Yong Li, Juan Li, Xiao-Ming Li, Qing-Qiang Kong, Cheng-Meng Chen

**Affiliations:** 1Key Laboratory of Materials for Energy Conversion and Storage of Shanxi Province, Institute of Molecular Science, Shanxi University, Taiyuan 030006, China; 15535437139@163.com (Q.Z.); zhaoyun@sxu.edu.cn (Y.Z.); 2Department of Materials Science and Engineering, Jinzhong University, Jinzhong 030619, China; snj642370134@126.com; 3Research Center for Fine Chemicals Engineering, Shanxi University, Taiyuan 030006, China; liyong@sxu.edu.cn; 4Institute of Crystalline Materials, Shanxi University, Taiyuan 030006, China; lj0511@sxu.edu.cn; 5CAS Key Laboratory of Carbon Materials, Institute of Coal Chemistry, Chinese Academy of Sciences, Taiyuan 030001, China; lixiaoming@sxicc.ac.cn (X.-M.L.); kongqq@sxicc.ac.cn (Q.-Q.K.); ccm@sxicc.ac.cn (C.-M.C.)

**Keywords:** silicon nanosheet, nitrogen–doped graphene, catalytic pyrolysis, embedded structure, lithium–ion battery

## Abstract

Silicon–based composites are promising candidates as the next–generation anode materials for high–performance lithium–ion batteries (LIBs) due to their high theoretical specific capacity, abundant reserves, and reliable security. However, expensive raw materials and complicated preparation processes give silicon carbon anode a high price and poor batch stability, which become a stumbling block to its large–scale practical application. In this work, a novel ball milling–catalytic pyrolysis method is developed to fabricate a silicon nanosheet@amorphous carbon/N–doped graphene (Si–NSs@C/NG) composite with cheap high–purity micron–size silica powder and melamine as raw materials. Through systematic characterizations such as XRD, Raman, SEM, TEM and XPS, the formation process of NG and a Si–NSs@C/NG composite is graphically demonstrated. Si–NSs@C is uniformly intercalated between NG nanosheets, and these two kinds of two–dimensional (2D) materials are combined in a surface–to–surface manner, which immensely buffers the stress changes caused by volume expansion and contraction of Si–NSs. Attributed to the excellent electrical conductivity of graphene layer and the coating layer, the initial reversible specific capacity of Si–NSs@C/NG is 807.9 mAh g^−1^ at 200 mA g^−1^, with a capacity retention rate of 81% in 120 cycles, exhibiting great potential for application as an anode material for LIBs. More importantly, the simple and effective process and cheap precursors could greatly reduce the production cost and promote the commercialization of silicon/carbon composites.

## 1. Introduction

In recent years, with the rapid development of artificial intelligence, big data, 5G communication, and electric vehicles, the booming demand for high–performance LIBs has attracted much attention from scientists around the world [1]. Graphite is the main commercialized anode material for LIBs; however, its low theoretical specific capacity (372 mAh g^−1^) and poor rate capability have limited the further application of LIBs [2]. Silicon is widely regarded as a next–generation anode material due to its high theoretical specific capacity (4200 mAh g^−1^), abundant natural reserves (26.4%), and safe voltage platform (0.4~0.5 V vs. Li^+^/Li) [3]. However, silicon–based anode materials face problems including physical/chemical property defects, complex production processes, and high costs. The great volume expansion and contraction (300~400%) of silicon during the lithiation/delithiation process causes structural crushing of the silicon–based material and further results in a rapid decay of the specific capacity [4]. What is more, the inherently poor conductivity and ion diffusion coefficient of silicon generate the sluggish electrode kinetics [5].

To overcome the above–mentioned defects, silicon/carbon (Si/C) composite anode materials have become a research hotspot [6]. According to the different crystal structure of silicon, amorphous silicon/carbon and crystalline silicon/carbon composites are the research hotspot at present. Due to the high lithium potential and good intrinsic strain/stress tolerance of amorphous silicon [7,8], amorphous silicon/carbon is an important branch. For example, the Kumar group designed a nano–architectured composite anode composed of active amorphous silicon domains (a–Si, 20 nm) and crystalline iron disilicide (c–FeSi_2_, 5~15 nm) alloyed particles dispersed in a graphite matrix realizing long–term mechanical, structural, and cycling stability [9]. Li et al. synthesized an amorphous–Si@SiOx/Cr/carbon (a–Si@SiOx/Cr/C) anode composite for lithium storage, exhibiting a stable discharge capacity of approximately 810 mAh g^−1^, with good capacity retention up to 200 cycles [10].

At present, crystalline silicon/carbon is still the dominant product and its microstructure types mainly include a core–shell structure, yolk–shell structure, a porous structure, and an embedded structure [11]. Among these Si/C composites with different structures, carbon material acting as a conductive matrix and a buffer medium could provide the buffer space for the volume change in nano–silicon and increase the electrical conductivity of the composite [12]. Graphene has become a star matrix material due to its advantages in terms of structure, chemical stability, flexibility, and ion/electron mobility [13,14]. Combining graphene with nano–silicon, the reversible capacity, cycling performance, and rate capability of Si/C composites could be effectively enhanced [15]. For example, Tang and co–workers synthesized a three–dimensional N–doped graphene/silicon composite, and the reversible capacity of electrode was 1132 mAh g^−1^ after 100 cycles at 5 A g^−1^ [16]. Liu and his colleagues synthesized a micron–sized Si@N–doped graphene composite, and the specific capacity of electrode was 518 mAh g^−1^ after 80 cycles at 500 mA g^−1^ [17]. Na et al. prepared a Nano–Si@N–doped graphene composite, and the electrode displayed a reversible specific capacity of 522 mAh g^−1^ at 2 A g^−1^ after 200 cycles [18]. Therefore, silicon/graphene composites, as a representative of Si/C composites, have the advantages of high capacity, excellent rate performance, and long cycle performance. However, hampered by the expensive raw materials and complex preparation process, they are confronted with difficulties in scale promotion.

Therefore, exploring simple, controllable, and low–cost preparation methods for graphene, nano–silicon, and their composites has become a research bottleneck of silicon/graphene composites towards practical application. At present, the redox method, the physical stripping method, and the chemical vapor deposition method could realize large–scale industrial production of graphene, but with challenges including a complex preparation process, environmental pollution, a low graphene monolayer rate, and the high cost of production [19,20,21]. Of concern is that several reports in recent years on nitrogen–doped graphene (NG), which is fabricated from graphitic carbon nitride (g–C_3_N_4_) in the heat treatment process due to the similar planar 2D sheet structure of g–C_3_N_4_ as graphene, have demonstrated a feasible preparation route of graphene [22,23,24,25,26].

On the other hand, bottom–up approaches of nano–silicon materials (nanospheres, nanowires, arrays, quantum dots, etc.) have greatly promoted the studies of Si/C materials. Compared with 0D and 1D silicon nanoelectrodes, 2D Si nanosheets (Si–NSs) have a higher specific surface area and stress relaxation to exhibit more stable cycling performance and a higher capacity retention rate [27]. However, at present, the preparation of silicon nanosheets mostly requires magnesium heat, a pickling process or chemical vapor deposition, which is difficult to use on a large scale due to its complexity and high cost.

It is conceivable that the silicon nanosheet/graphene composites prepared by low–cost raw materials and a simple preparation process are expected to become widely applied high–performance anode materials. Unfortunately, no similar studies have been reported so far. In order to fabricate low–cost silicon nanosheet/graphene composites by a simple preparation process, we developed a novel ball milling–catalytic pyrolysis method with melamine and micron silicon powders as precursors and produced an embedded structured Si–NSs@C/NG. After mixing g–C_3_N_4_ with silicon nanosheets by ball milling, the conversion of g–C_3_N_4_ to NG and the carbon coating of silicon nanosheets were realized in one step when the mixture was catalyzed by ferrocene. As an anode material of LIBs, Si–NSs@C/NG exhibited a good electrochemical performance and the reversible specific capacity was 807.94 mAh g^−1^ at 200 mA g^−1^, with a capacity retention rate of 81% in 120 cycles. This work has effectively reduced the cost and simplified the preparation process of silicon/graphene composite materials.

## 2. Results and Discussion

### 2.1. Synthesis and Microstructure Characterization

The morphology and microstructures of samples are shown in Figure 1. Obviously, Si–NSs show a typical nanosheet morphology (Figure 1a) with a thickness of approximately 75~85 nm (Appendix A) and the lateral size concentrated at 126 nm (Appendix A). Figure 1b exhibits the SEM image of NG synthesized by directly mixing g–C_3_N_4_ and ferrocene followed by heat treatment, which presents many obvious lamellar structures. Figure 1c,d and Appendix A present the morphologies of Si–NSs@C/NG, in which the Si–NSs are uniformly anchored on the NG nanosheets. NG in the composite material forms a stable conductive network, which is not only conducive to the uniform dispersion of Si–NSs and the alleviation of the volume change in silicon, but also provides storage space for the electrolyte, which is beneficial for the rapid transfer of electrons and ions. Figure 1e displays that Si–NSs are uniformly anchored or embedded in the wrinkled NG network. The HRTEM image shows that Si–NSs maintain their crystal structure, with a lattice spacing of 0.31 nm, corresponding to the Si (111) crystal plane (Figure 1f). At the same time, a large number of nanoparticles with a lattice spacing of 0.25 nm belonging to the Fe_2_O_3_ (311) crystal plane could be found, indicating that Fe_2_O_3_ nanoparticles are generated and embedded on the NG sheets. Meanwhile, Si–NSs and Fe_2_O_3_ are wrapped by an amorphous carbon layer, with a thickness of approximately 2 nm, indicating that Fe_2_O_3_@C and Si–NSs@C are formed simultaneously with the forming process of NG. EDX mapping demonstrates that the distribution of C, Fe, N, Si, and O is uniform (Appendix A). It manifests that the graphene formed is doped with plentiful nitrogen, which is consistent with the related studies [23,24,25,26]. It also suggests that Si–NSs could be dispersed evenly in NG by a ball milling–catalytic pyrolysis method.

Figure 2a shows the XRD patterns of Si–NSs, NG, and Si–NSs@C/NG. The characteristic peaks of silicon (JCPDS No. 27-1402) located at 28.4°, 47.3°, 56.1°, 69.1°, and 76.4° are obvious, indicating that the crystal structure of Si–NSs prepared by sand milling is complete. For NG, the carbon peak located at 23.5° is a broad camel peak (Appendix A), indicating that NG crystal formed by g–C_3_N_4_ (catalyzed by ferrocene) has many structural defects and a large layer spacing [25]. However, the characteristic peaks of Fe_2_O_3_ (JCPDS No. 39-1346) appearing at 30.2°, 35.6°, 43.2°, 53.7°, 57.2°, and 62.9° in NG is significant, suggesting the high crystallinity of Fe_2_O_3_ generated during the catalytic pyrolysis process. For Si–NSs@C/NG, there are typical characteristic peaks of silicon, but the characteristic peaks of C and Fe_2_O_3_ are not obvious, which may be related to the low content of Fe_2_O_3_ in the composite and plenty of defects in NG.

The specific surface area and pore structure of the samples are measured by the nitrogen adsorption–desorption method, as shown in Figure 2b. NG and Si–NSs@C/NG exhibit typical type Ⅳ isothermal curves with H3 hysteresis loops, revealing that both materials have sheet–like structures with mesopores. The BET specific surface areas of NG and Si–NSs@C/NG are 286.2 m^2^ g^−1^ and 37.6 m^2^ g^−1^, respectively. From the inset of Figure 2b, it could be discovered that the pore size distribution of Si–NSs@C/NG is relatively wide. Such a porous structure could buffer the huge volume change in silicon during lithiation/lithiation and improve the contact between the electrolyte and the electrode material, which promote the rate capability of the batteries [28].

Figure 2c shows the TGA of NG and Si–NSs@C/NG in the air atmosphere. Before 450 °C, the mass of both decayed rapidly, indicating that there are lots of carbon components, which is consist with TEM and XRD results. After 450 °C, the mass remains stable and increases slightly, and the final yields of NG and Si–NSs@C/NG are 57.52% and 56.94%, respectively. Combining the material composition, the final residue of NG should be Fe_2_O_3_, and that of Si–NSs@C/NG is SiO_2_ and Fe_2_O_3_. The content of Fe_2_O_3_ in NG reaches 57.52%, confirming the phenomenon that a large number of Fe_2_O_3_ nanoparticles are dispersed on NG nanosheets in Appendix A. ICP could more accurately determine the content of Si and Fe in Si–NSs@C/NG, and the results are 36.30% and 4.32%, respectively. It can be further calculated that the contents of Fe_2_O_3_ and Si–NSs in the composite are 6.17% and 36.3%, respectively. Therefore, the total content of carbon and nitrogen in Si–NSs@C/NG is 57.53%, much higher than the total content of carbon and nitrogen in NG (42.48%). What is more, the yields of (NG+C+Fe_2_O_3_) in NG and Si–NSs@C/NG are 0.99% and 6.02%, respectively, which also indicates that the addition of Si–NSs improves the yield of NG and coated carbon. The possible reason is that Si–NSs dispersed on the g–C_3_N_4_ lamellae increase the deposition area of carbon molecules generated by the decomposition of ferrocene, which is equivalent to expanding the molecular weight of g–C_3_N_4_, hindering its decomposition and increasing the composite yield. This is consistent with the fact that the specific surface area of NG is much higher that of the composite material.

Figure 2d shows the Raman spectra of NG and Si–NSs@C/NG, and the peaks at 287.3, 498.7, and 916.8 cm^−1^ are characteristic peaks of crystalline silicon. The peaks at 1371 and 1593 cm^−1^ correspond to the D peak and G peak of NG [29]. The appearance of the D peak is due to the synthesis of porous graphene with abundant structural defects caused by N doping, and the G peak represents the sp^2^–hybridized carbon network structure [30]. At the same time, the basic Raman scattering peaks of Fe_2_O_3_ could also be observed in the Si–NSs@C/NG spectrum, corresponding to the A_1g_ mode at 220 cm^−1^, the A_g1_ mode at 594 cm^−1^, and the E_g_ mode at 414 cm^−1^ [31]. However, the peak of Fe_2_O_3_ in NG is not obvious, which may be caused by the surface amorphous carbon coating. The intensity ratio of the D–band and G–band (*I_D_/I_G_*) of NG and Si–NSs@C/NG is 1.14 and 0.92, respectively, indicating more defects of NG than that of Si–NSs@C/NG, which also confirms that the presence of Si–NSs is beneficial to synthesize NG and reduce defects.

The elemental information of Si–NSs@C/NG is analyzed by XPS. As shown in Figure 3a, the full spectrum of Si–NSs@C/NG contains O1s, C1s, N1s, Si2p, and Fe2p. Appendix A reveals that the Si content on the surface of the composite is only 6.52%, which is much lower than that in the bulk phase of the material. This indicates that Si–NSs are effectively coated by carbon layer and embedded inside the NG, rather than only barely dispersed on the surface of graphene. Figure 3b shows the Fe2p spectrum of Si–NSs@C/NG in which two different peaks could be observed, corresponding to the spin–orbit logarithms of Fe2P_3/2_ (710.9 eV) and Fe2p_1/2_ (724.2 eV). In addition, these two peaks are accompanied by two satellite peaks located at 718.6 and 733.3 eV, which represent the Fe(III) valence state [32]. This again shows that the iron in the composite is finally in the form of Fe_2_O_3_, which is consistent with the analysis results of TEM, XRD and Raman. Therefore, the existence of oxygen in the composite is mainly derived from Fe_2_O_3_. Figure 3c is the C1s spectrum of Si–NSs@C/NG, and the peaks at 284 and 284.4 eV correspond to C=C and C–C bonds, which are consistent with the chemical bond type of graphene. The peaks at 285.5 and 286.7 eV correspond to C=N and C–N bonds, which indicate that N is successfully doped into the graphene structure, and the peaks at 286.2 and 288.5 eV correspond to C–O–C and O–C=O bonds [33]. Furthermore, it can also be seen from the figure that the peak intensities of C=C and C–C bonds are significantly higher than those of C=N and C–N bonds, which indicates that only a small amount of C atoms in the chemical bonds of graphene are replaced by N atoms. Figure 3d shows the N1s spectrum of the composite. The three peaks at 398.3, 399.6, and 401.1 eV correspond to pyridine N, pyrrolic N, and graphitic N, respectively [34]. The N content in the surface of the composite is 11.64%, and the presence of N can bring more defects to the composite, thereby providing more active sites for lithium ion storage [33].

According to the above analysis, the preparation process of Si–NSs@C/NG could be concluded and mainly divided into two stages. As shown in Figure 4, firstly, when g–C_3_N_4_ is mixed with Si–NSs in the ball mill tank, Si–NSs can be uniformly embedded in the g–C_3_N_4_ sheets. Secondly, in a closed quartz tank, ferrocene initially sublimates and decomposes into gaseous products (Fe, H_2_, CH_4_, C_2_H_6_, etc.) above 550 °C. The –NH_2_/–NH groups of g–C_3_N_4_ will also decompose from the skeleton into the atmosphere of the quartz tank [22]. At the same time, the clusters of iron atoms obtained from the decomposition of ferrocene are aggregated to form nanoparticles, which will sink into the skeleton structure of g–C_3_N_4_ [23]. When the temperature rises to 700 °C, the skeleton structure of g–C_3_N_4_ changes to the graphene structure, with a large number of holes [24]. At this time, iron nanoparticles could catalyze the hydrocarbon molecules in the quartz tank to form the graphene structure, which repair the holes to make the structure more complete. Meanwhile, the nitrogen–containing molecules in the quartz tank will recombine with the carbon atoms at the defect of the graphene structure with high activity to form a nitrogen–doped configuration dominated by pyridine nitrogen and pyrrole nitrogen [26]. Some hydrocarbon molecules are deposited on the surface of exposed Si–NSs as carbon sources to form carbon coatings. Finally, oxygen molecules in the quartz tank will combine with highly active iron nanoparticles to form Fe_2_O_3_ nanoparticles, and some hydrocarbon molecules are deposited on the surface of Fe_2_O_3_ nanoparticles to form amorphous carbon coating. In this way, the Si–NSs@C/NG composite is successfully prepared.

### 2.2. Electrochemical Performances

Figure 5a shows the cyclic voltammetry of Si–NSs@C/NG. A broad and large reduction peak appears at 0.6 V during the first cycle of lithium intercalation, which corresponds to the formation of Solid electrolyte interphase (SEI) films [35]. This peak disappears in the following cycle, indicating that the resulting SEI film is relatively complete and stable. There is a reduction peak at 0.15 V during the subsequent cycle discharge, which gradually increases with the number of cycles, corresponding to the formation of Si–Li alloys [36]. During the delithiation process, gradually enhanced oxidation peaks appear at 0.35 V and 0.53 V, which corresponds to the dealloying process [37]. The ion diffusion kinetics and charge transfer of the materials are further explored by the EIS. Figure 5b shows the Nyquist plot and corresponding equivalent circuit model of Si–NSs, NG, and Si–NSs@C/NG. Compared with Si–NSs and NG, Si–NSs@C/NG has smaller charge transfer resistance and ion diffusion resistance. NG with network structure could provide good electron conduction channels and effectively promote charge transport. The effective intercalation of Si–NSs and NG in the composite could better inhibit the particle pulverization and detachment caused by volume expansion when lithium ions are intercalated and deintercalated, which is conducive to the diffusion of lithium ions.

In order to further study the microdynamics of electrode reaction, GITT (Figure 6a) is used to calculate the diffusion coefficient of lithium ions (D_Li_^+^) [38,39]. As shown in Figure 6b, Si–NSs@C/NG exhibits the highest D_Li_^+^ value for both the discharge and charge processes, which is conducive to the excellent rate capability of the electrode [40,41,42]. Furthermore, the diffusion coefficients of NG are lower than those of Si–NSs, because NG contains a large amount of Fe_2_O_3_, which will hinder the diffusion of lithium ions. Figure 6c shows the D_Li_^+^ of active electrodes at different voltages, Si–NSs@C/NG also has the highest D_Li_^+^ value, which is consistent with the results of the EIS test.

Figure 7a shows the first charge–discharge curves of Si–NSs, NG, and Si–NSs@C/NG with the initial Coulombic efficiency (ICE) of 20.96%, 17.73%, and 42.45%, respectively. Although the ICE of Si–NSs@C/NG is much higher than that of Si–NSs and NG, it is still relatively low. The reason may be mainly attributed to that the large specific surface area consumes a large amount of lithium ions to form a stable SEI, and the large number of pores and defects also increase abundant side reactions, resulting in the gratuitous consumption of lithium ions. At the same time, the selected charging voltage range (0.01~1.5 V) also causes that part of the lithium ions embedded in the material are not embedded out, resulting in low ICE. Figure 7b presents representative galvanostatic charge/discharge curves of Si–NSs@C/NG at 5th, 20th, 50th, 100th, and 120th at 200 mA g^−1^. The composite exhibits a plateau at approximately 0.15 V, which corresponds to the formation of the amorphous Li_x_Si phase. It also has a typical silicon charging plateau at approximately 0.5 V, which is consistent with the results of the CV test.

As shown in Figure 7c, compared to the other two electrode materials, Si–NSs@C/NG exhibits a higher reversible capacity and better cycling stability. At a current density of 200 mA g^−1^, the initial charge specific capacity of Si–NSs@C/NG is 807.94 mAh g^−1^, and the reversible specific capacity can still be maintained at 654.32 mAh g^−1^ after 120 charge–discharge cycles, with a capacity retention rate of 81%. While the reversible specific capacity of Si–NSs and NG decay rapidly at the beginning of the first cycle. Furthermore, compared with the currently reported silicon/N–doped graphene composites as anode materials for LIBs (Table 1), Si–NSs@C/NG electrode manifests certain advantages in the capacity retention rate. What is exciting is that the method introduced in this work is the simplest, and the cost is also the lowest, which could facilitate the commercialization of the Si/C anode.

Figure 7d displays the rate performance of Si–NSs, NG, and Si–NSs@C/NG. At current densities of 50, 100, 200, 400, 800, and 1600 mA g^−1^, Si–NSs@C/NG achieves 809.37, 772.51, 720.43, 638.14, 503.77, and 294.55 mAh g^−1^, respectively. When the current density is restored to 100 mA g^−1^, the reversible specific capacity could be returned to 83.5% of the previous value at the same current density. Si–NSs and NG perform very poorly under the same test condition, which demonstrates the effective intercalation of NG and Si–NSs as well as the carbon coating layer in the composite, which increases the overall stability of the material.

The Si–NSs@C/NG electrode after 120 cycles is disassembled and rinsed with dimethyl carbonate. The SEM and TEM images of the electrode are shown in Figure 8. It can be observed that the structure of the electrode remains intact without excessive pulverization and fracture, reconfirming the structural stability of the Si–NSs@C/NG. Some inconspicuous cracks are unavoidable in the electrode which cannot cause electrode degradation. More importantly, the Si–NSs@C/NG electrode still remains embedded structure (Figure 8c,d), and the stable material structure results in excellent electrochemical performances.

Combined with the microstructure of Si–NSs@C/NG, SEM and TEM data of the electrodes after 120 cycles, and electrochemical performance, it can be seen that the massive g–C_3_N_4_ was separated into sheets due to ball milling and bonded together to form a large number of pores. The porous NG was formed through the pyrolytic catalytic reaction of ferrocene with g–C_3_N_4_. The process of ball milling and catalytic pyrolysis dispersed the silicon nanosheets on the surface and between the layers of the nitrogen–doped graphene, and the amorphous carbon coating was formed in situ on the surface to achieve the stable binding of the silicon nanosheets and graphene. It was observed that the lamellar structure of the composite remained after 120 charge–discharge cycles, while the silicon nanosheets retained their original shape and could still be anchored to the graphene sheets. Thus, it could be imagined that lithium ions are diffused to nitrogen–doped graphene and silicon nanosheets through the liquid phase and pores, embedded in graphene to form Li_2_C_x_ (x < 6) and Li–Si alloy with silicon nanosheets to form Li_22_Si_5_ at most [23,25]. Due to the excellent conductive network of nitrogen–doped graphene, 2D structure of silicon nanosheets and amorphous carbon coating on silicon surface, the huge stress caused by volume expansion–contraction is greatly buffered and structural damage is avoided. The structural stability of the material is preserved to the greatest extent, and the high cyclic stability and the high capacity retention rate of the material are also realized. The excellent electrochemical performance of Si–NSs@C/NG could be attributed to the following points: (1) Si–NSs are embedded in NG with lamellar structure in sheet form, and the two 2D materials are combined in a face–to–face manner, which improves the overall stability of the material; (2) NG could greatly buffer the stress changes caused by the volume expansion–contraction of Si–NSs; (3) both the amorphous carbon coating layer and the graphene layer have excellent electrical conductivity, which is beneficial to the transport of electrons and lithium ions; (4) N doping could provide more active sites, which is helpful for the composite to obtain better performance during cycling.

## 3. Conclusions

In summary, a novel ball milling–catalytic pyrolysis method was developed to prepare an innovative intercalated structured carbon–coated silicon nanosheet/N–doped graphene composite (Si–NSs@C/NG). As an anode material for LIBs, Si–NSs@C/NG exhibits excellent electrochemical performance, with an initial reversible capacity of 807.94 mAh g^−1^, and with a capacity retention rate of 81% after 120 cycles. These excellent electrochemical performances originate from the unique structural design of Si–NSs@C/NG, in which Si–NSs could effectively intercalate into NG and are protected by a pyrolytic amorphous carbon coating. The silicon nanosheets in the composite material are made by the sanding method using cheap high–purity silicon micron powder as the raw material, while the raw material nitrogen–doped graphene is fabricated from melamine. Further, the adopted ball milling–catalytic pyrolysis method is conducive to simplifying the production process and reducing the production cost of Si/C composites, suggesting the great potential for commercial application.

## 4. Materials and Methods

### 4.1. Materials Preparation

#### 4.1.1. Preparation of g–C_3_N_4_ and the Si–NS Slurry

g–C_3_N_4_ was prepared by a thermal polymerization method. Briefly, melamine (10.0 g) was added to a corundum crucible with a lid, and was then heated up to 550 °C, with a heating rate of 5 °C min^−1^, and kept for 4 h in a muffle furnace under static air. After being cooled naturally to room temperature, a yellow bulk sample was grounded, collected, and labeled as g–C_3_N_4_. The Si–NS slurry was obtained by sand grinding high–purity micron silicon powders (99.9%) with ethanol as the solvent. The grinding was carried out for 2 h in the SMLYU ultra–fine sand grinder using zirconia balls, with a size of 0.3/0.6 mm, followed by an additional 4 h of grinding with 0.1 mm zirconia balls in the SMLV ultra–fine sand grinder.

#### 4.1.2. Preparation of Si–NSs@C/NG

The Si–NS slurry (0.32 g mL^−1^) containing 0.1 g Si–NSs and 3.0 g g–C_3_N_4_ was dispersed in 20 mL anhydrous ethanol solution, and then poured into the ball–milling jar to grind at a speed of 400 r min^−1^ for 1 h (rotate forward and reverse for 30 min, respectively) with a ball–material ratio of 15:1. The obtained slurry was heated and evaporated by stirring to obtain a brown powder. After that, 3.1 g brown powder and 0.2 g ferrocene were mixed evenly in a mortar and sealed in a 90 mL quartz tank. Then, the mixture was put it into a muffle furnace filled with air atmosphere, and the temperature was kept constant at 700 °C for 2 h, with a heating rate of 3 °C min^−1^. Finally, 0.2926 g Si–NSs@C/NG was collected after cooling to room temperature. NG was prepared by weighing 3.0 g g–C_3_N_4_ and 0.2 g ferrocene under the same heating conditions as above, and then 0.0318 g of sample was obtained. Appendix A provides the detailed calculation process of the yield of (NG+C+Fe_2_O_3_) in products.

### 4.2. Material Characterization

The crystal structure, morphology and microstructure of the samples were characterized by X–ray diffraction (XRD, D8 Advance, Bruker, Germany), Raman spectroscopy (Raman, LabRAM HR Evolution, Horiba, Japan), scanning electron microscopy (SEM, JSM-7001F, JEOL, Japan), an energy–dispersive X–ray spectrometer (EDX), a transmission electron microscope (TEM, JEM-2800, JEOL, Japan), and X–ray photoelectron spectroscopy (XPS, Scientific K–Alpha, Thermo, Waltham, MA, USA). The Si–NSs content of the samples was measured by thermogravimetric analysis (TGA, 7300, Hitachi, Japan) in the air atmosphere and inductively coupled plasma spectrometer (ICP, 5110, Agilent, Santa Clara, CA, USA). The specific surface area and pore size distribution of the samples were tested by a specific surface area analyzer (BET, ASAP2460, Micromeritics, Norcross, GA, USA).

### 4.3. Electrochemical Measurement

The as–prepared active substances, carbon nanotubes, and sodium alginate were mixed at a mass ratio of 85:9:6, deionized water was added drop wise, and the mixture was ground into evenly mixed slurry and coated on copper foil, then dried at 120 °C in a vacuum drying oven for 12 h. The copper foil was cut into a 12–mm diameter electrode sheet, with a microtome as the working electrode, and the lithium sheet was applied as the counter electrode and the reference electrode. The electrolyte was LiPF_6_ (1 M) in ethylene carbonate and dimethyl carbonate (EC/DMC) at a 1:1 (*v*/*v*) volume ratio with 10 wt% fluoroethylene carbonate (FEC). CR2016 half–cells were assembled in an Ar–filled glove box (Jiumen, Nanjing, China) with a Celgard–2400 membrane as the separator. The active material loading of the working electrode was 1.0~1.2 mg cm^−2^. Cyclic voltammetry (CV) tests were performed on a CHI 660C electrochemical workstation (Shanghai Chenhua Company, Shanghai, China) with a voltage range of 0.01–1.5 V (vs. Li^+^/Li) at a scanning rate of 0.25 mV s^−1^. The electrochemical impedance spectrum (EIS) was measured at a frequency range from 10 MHz to 0.01 Hz. The galvanostatic charge–discharge tests were performed at room temperature using a BTSCT-4008 system (Neware, Shenzhen, China), with a voltage range of 0.01~1.5 V (vs. Li^+^/Li). The diffusion coefficient of lithium ions was measured by a galvanostatic intermittent titration technique (GITT), and the batteries were (dis)charged at 100 mA g^−1^ for 25 min and maintained in an open circuit for 2 h.

## Figures and Tables

**Figure 1 molecules-28-03458-f001:**
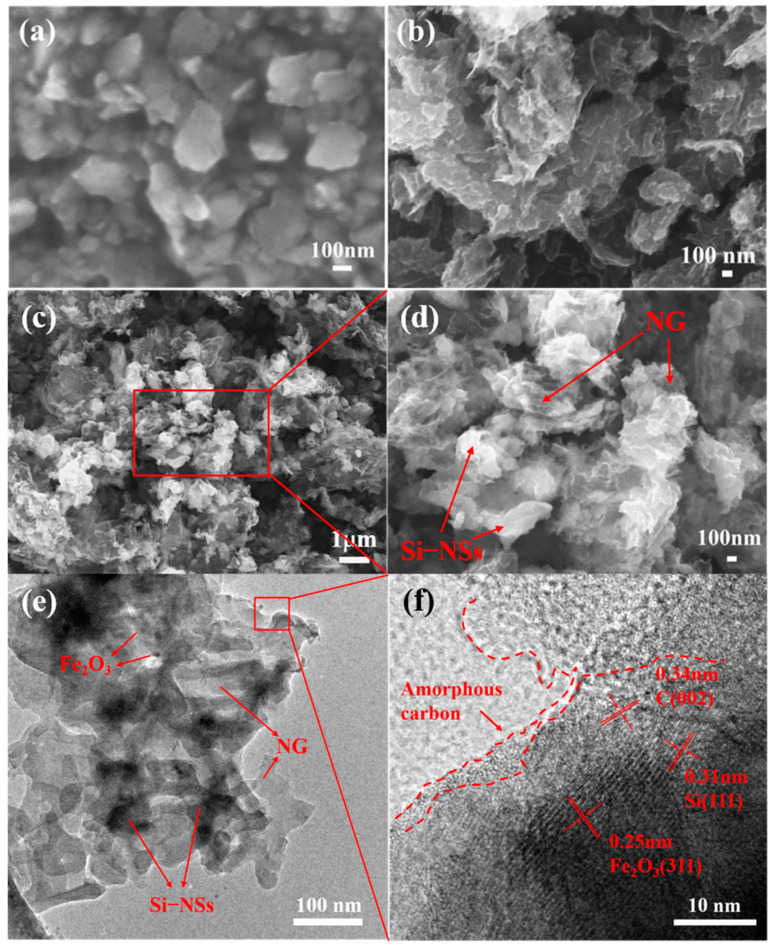
SEM images of Si–NSs (**a**); NG (**b**); Si–NSs@C/NG (**c**,**d**); TEM image (**e**) and HRTEM image (**f**) of Si–NSs@C/NG.

**Figure 2 molecules-28-03458-f002:**
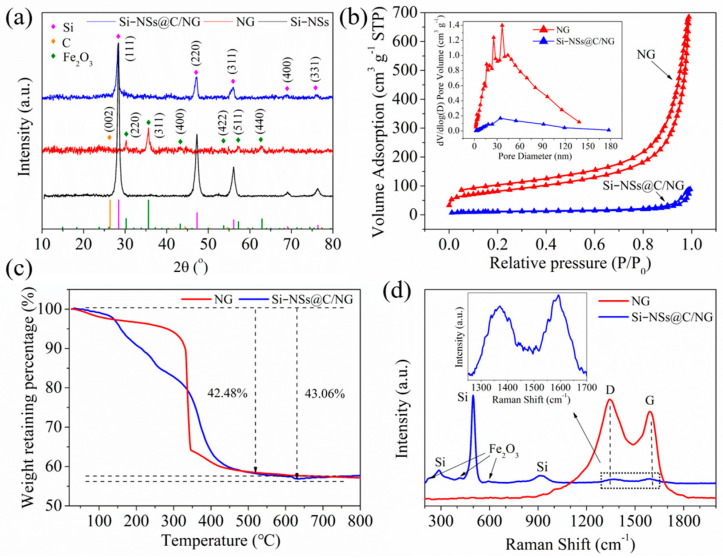
(**a**) XRD patterns of Si–NSs, NG, and Si–NSs@C/NG; (**b**) nitrogen adsorption–desorption isotherms of NG and Si–NSs@C/NG along with the pore size distribution (inset); TGA curves (**c**) and Raman spectra (**d**) of NG and Si–NSs@C/NG.

**Figure 3 molecules-28-03458-f003:**
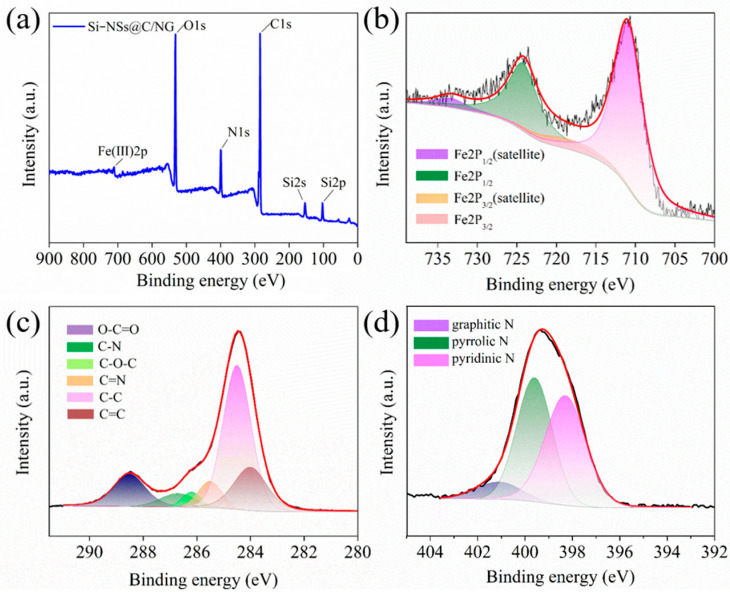
(**a**) XPS survey spectra of Si–NSs@C/NG; spectrum of (**b**) Fe2p; (**c**) C1s; (**d**) N1s.

**Figure 4 molecules-28-03458-f004:**
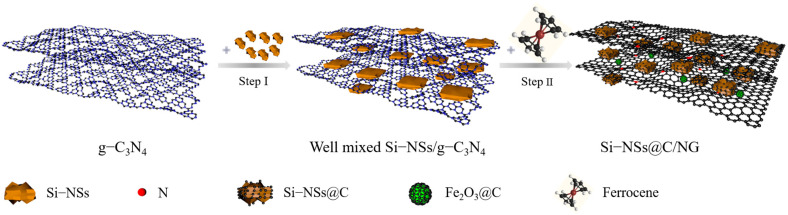
Schematic illustration of the synthesis process of Si–NSs@C/NG.

**Figure 5 molecules-28-03458-f005:**
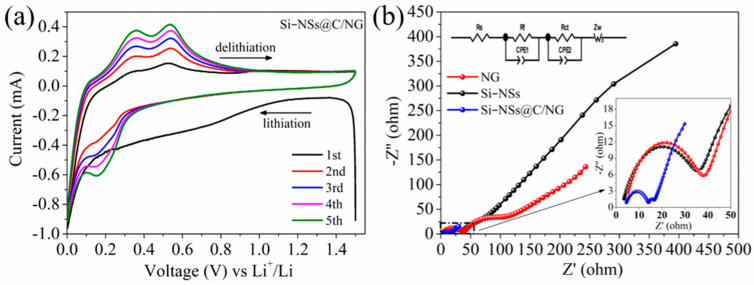
(**a**) CV curves of Si–NSs@C/NG at a scan rate of 0.25 mV s^−1^; (**b**) Nyquist plot and corresponding equivalent circuit model of Si–NSs, NG, and Si–NSs@C/NG.

**Figure 6 molecules-28-03458-f006:**
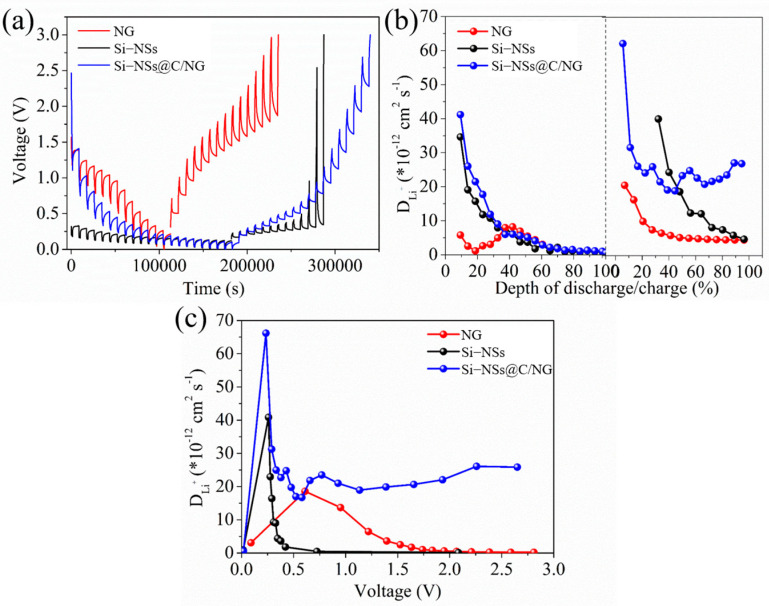
(**a**) GITT curves of Si–NSs, NG and Si–NSs@C/NG electrodes and (**b**) corresponding lithium ion diffusion coefficients; (**c**) lithium ion diffusion coefficients of active electrodes at different voltages.

**Figure 7 molecules-28-03458-f007:**
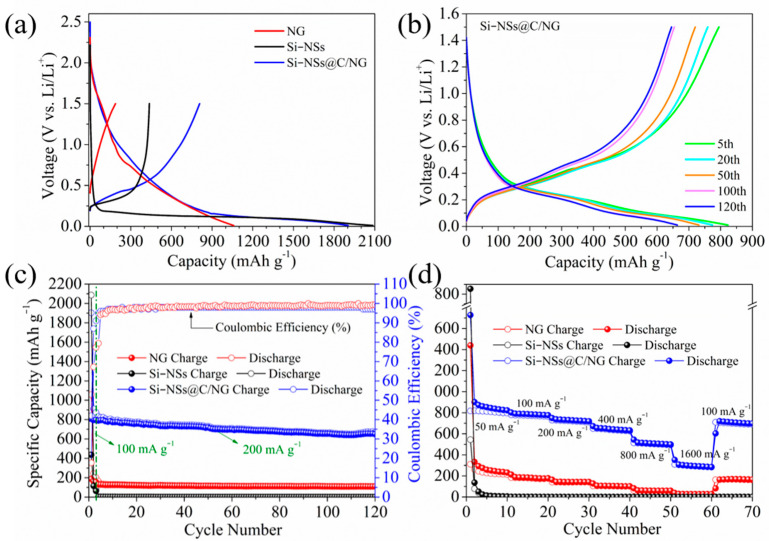
(**a**) Galvanostatic charge/discharge curves for the first cycle of Si–NSs, NG and Si–NSs@C/NG; (**b**) galvanostatic charge/discharge curves of Si–NSs@C/NG at 5th, 20th, 50th,100th, and 120th at 200 mA g^−1^; (**c**) cycling performance of Si–NSs, NG, and Si–NSs@C/NG at 200 mA g^−1^; (**d**) rate performance of Si–NSs, NG, and Si–NSs@C/NG.

**Figure 8 molecules-28-03458-f008:**
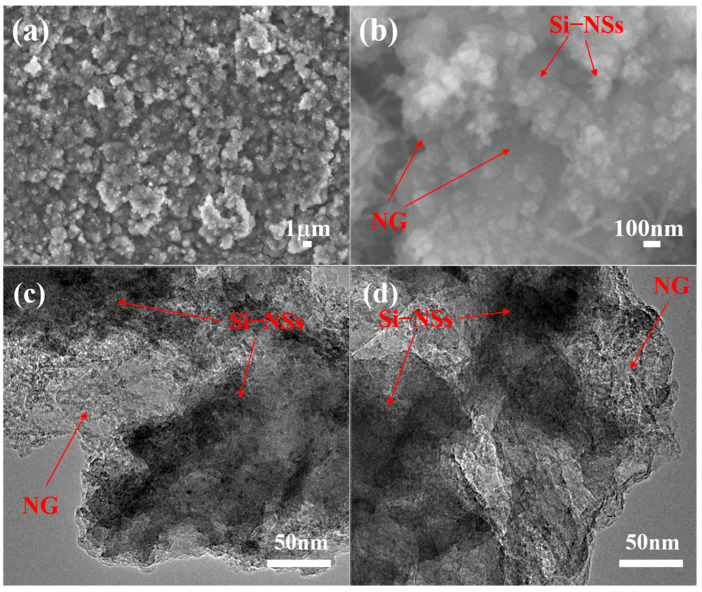
(**a**,**b**) SEM images of Si–NSs@C/NG electrode after 120 cycles; (**c**,**d**) TEM images of Si–NSs@C/NG electrode after 120 cycles.

**Table 1 molecules-28-03458-t001:** Comparisons of the electrochemical performance of Si–NSs@C/NG with other silicon/N–doped graphene anodes in Li–ion half cells.

Electrochemical Properties				
Anodes Material /Ref.	Preparation Methods	Current Density (mAg^−1^)	Specific Capacity (mAhg^−1^) /Cycle Number	Capacity Retention Rate (%)
3DNG–P@Si [16]	improved Hummers method and electrostatic self–assembly	1000	1017/200	75
PSNGM [43]	spray drying	100	1142/150	96.1
NG/Si@NC [44]	solution–mixing and carbonization	500	938/100	82
Nano–Si@NG [18]	modified Hummers method	2000	552/200	53
Si/NG [45]	carbonization and low–temperature chemical reduction	1000	1138/240	42
G–Si [46]	carbon–thermal method	200	819/50	62
Si/NC/NG [47]	solution–mixing and carbonization method	500	1210/100	58
mSi@GNG [17]	modified Hummers method and freeze–drying	500	510/80	58
Si@NC/G [48]	drying and high–temperature calcination	100	1045/200	56
Si–NSs@C/NG	ball milling–catalytic pyrolysis method	200	654/120	81

## Data Availability

Not applicable.

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
