# Peer review of "Constructing a Low–Cost Si–NSs@C/NG Composite by a Ball Milling–Catalytic Pyrolysis Method for Lithium Storage"

_molecules, 2023, doi:10.3390/molecules28083458_

Round 1

Reviewer 1 Report

This manuscript focuses on the “Constructing Low-cost Si-NSs@C/NG Composite by Ball milling-catalytic Pyrolysis for Lithium Storage”. The topic is of great relevance to scientific community as searching for alternative anodes materials is in high demand. The author’s highlights that the synthesis through ball milling catalytic pyrolysis is a low cost and scalable method, which is important for commercial and industrial applications. The overall design including synthesis, methodology and results are excellent and would be of importance for the readership of Molecules Journal.

While reviewing this article, I came across a similar paper that was submitted to SSRN Electronic Journal in January 2020 by the same authors. The details of the aforementioned article are given below:

“A Facile Ball Milling-Catalytic Pyrolysis Preparation of Si-Nss@C/Ng Composite for Lithium Storage”

January 2022, SSRN Electronic Journal, DOI: 10.2139/ssrn.4257573

Authors should clarify that they have published this article. Also, I found that they have used some results from the SSRN Electronic Journal paper, which is unethical and not acceptable in the scientific community without giving a proper citation. If the submitted manuscripts to Molecules is the extended version of the previous manuscript, then they must cite that article and clearly mentioned in the manuscript.

 I would not accept this manuscript in its current form and would recommend authors to follow the strict guidelines of the Journals.

Reviewer 2 Report

The authors reported the work entitled ”Constructing Low-cost Si-NSs@C/NG Composite by Ball mill-ing-catalytic Pyrolysis for Lithium Storage”. In this work, Si-NSs@C/NG) composite was prepared using cheap high purity micron-size silica powder and mela-mine as raw materials, where Si-NSs@C is uniformly intercalated between NG nanosheets. Si-NSs@C/NG Composite shows a reversible specific capacity of 807.9 mAh g−1 at 200 mA g−1 with a capacity retention rate of 81% in 120 cycles. This work provides an approach to synthesizing silicon/carbon composites via simple and effective process and cheap precursors that could reduce greatly the production cost. However, some critical issues should be resolved before its publication in Molecules.

1.       The reversible capacity of Si-NSs@C/NG Composite is much lower than those reported work in literature such as mentioned in the introduction part.

2.       It is written, “The excellent electrochemical performance and the suitable preparation method for large-scale production make the composite have the potential to be popularized.” However, the total products and yield are so low for large-scale production.

3.       It is still unclear how much Si and Fe2O3 are in the Si-NSs@C/NG Composite, respectively. Please clarify this in the manuscript.

4.       It is written, “Silicon is widely regarded as a next-gen anode material benefited from its high theoretical specific capacity (4200 mAh g−1), abundant natural reserves (26.4%), and safe voltage platform (0.4~0.5 V vs. Li+/Li)” and “It can be seen that the initial discharge capacity of the composite reaches 1900 mAh g−1, which is close to its theoretical capacity.” It should be mentioned how the theoretical capacity of Si-NSs@C/NG Composite was calculated. In addition, CV peak at 0.6 V during the first cycle of lithium intercalation was attributed to the formation of Solid electrolyte interphase (SEI) films, which contribute to a large portion of capacity and therefore disagree with the above-mentioned 1900 mAh g−1 of theoretical capacity.

Reviewer 3 Report

> what are the objectives of the present work? List out clearly in the introduction section.

> what are the gaps identified in the existing articles? how the proposed work is going to fullfill the gaps?

> Materials Preparation section requires more detail inputs to readers.

> 550 degree C requires a space.

> Preaparation techniques require elaborate details.

> Material Characterization: "The crystal structure, morphology and microstructure of the samples were characterized by ray diffraction" which ray diffraction?

> why the reference [24] is cited for figure 1(f)? is it reproduced?

> Figure 2 (a) which value is taken along x-axis? unit only given.

> Figure 3. Use units in curved bracket. avoid using /

> I could find lot of typoerrors and space corrections throughout the manuscript. 

> A thorough revision is required to polish the manuscript effectively.

Round 2

Reviewer 1 Report

The authors have clarified and took notice of the preprint version that was mistakenly posted on the SSRN. The overall manuscript is great for the readership of Molecules Journal.

I still feel that the introduction section could be improved. Especially, the authors correctly pointed out the issues (low capacity) of graphite followed by Si volume expansion and contraction that could lead to pulverization of Si, unstable formation of the SEI layer and hence the rapid capacity fading. These issues are standard with the Si/graphite composite system. However, in the last decade these has been several investigations (https://doi.org/10.1002/smll.201906812) on this topic by designing novel composite systems such as amorphous Si/graphite composite and these systems can provide unprecedented stability and better cycling performances. In my opinion, authors should not overlook these efforts and therefore a small paragraph must be included before accepting this manuscript for publications in Molecules.

Author Response

The authors have clarified and took notice of the preprint version that was mistakenly posted on the SSRN. The overall manuscript is great for the readership of Molecules Journal.

I still feel that the introduction section could be improved. Especially, the authors correctly pointed out the issues (low capacity) of graphite followed by Si volume expansion and contraction that could lead to pulverization of Si, unstable formation of the SEI layer and hence the rapid capacity fading. These issues are standard with the Si/graphite composite system. However, in the last decade these has been several investigations (https://doi.org/10.1002/smll.201906812) on this topic by designing novel composite systems such as amorphous Si/graphite composite and these systems can provide unprecedented stability and better cycling performances. In my opinion, authors should not overlook these efforts and therefore a small paragraph must be included before accepting this manuscript for publications in Molecules.

Respond:

We are very pleased with the reviewer's recognition.

We also agree with the importance of amorphous silicon/carbon system mentioned by the reviewer, and a paragraph is added in the paper to introduce the progress.

Revised part: [Page 2]

According to the different crystal structure of silicon, amorphous silicon/carbon and crystalline silicon/carbon composites are the research hotspot at present. Due to the high lithium potential and good intrinsic strain/stress tolerance of amorphous silicon [7, 8], amorphous silicon/carbon is an important branch. For example, Kumar group designed nano-architectured composite anode composed of active amorphous silicon domains (a-Si, 20 nm) and crystalline iron disilicide (c-FeSi2, 5~15 nm) alloyed particles dispersed in a graphite matrix realizes long-term mechanical, structural, and cycling stability [9]. Li et al synthesized amorphous-Si@SiOx/Cr/carbon (a-Si@SiOx/Cr/C) anode composite for lithium storage exhibitting a stable discharge capacity of about 810 mAh g-1 with good capacity retention up to 200 cycles [10].

Ref.:

  1. Lin, L.; Xu, X.; Chu, C.; Majeed, M. K.; Yang, J., Mesoporous Amorphous Silicon: A Simple Synthesis of a High-Rate and Long-Life Anode Material for Lithium-Ion Batteries. Angewandte Chemie International Edition 2016, 55, (45), 14063-14066.
  2. Orthner, H.; Wiggers, H.; Loewenich, M.; Kilian, S.; Bade, S.; Lyubina, J., Direct gas phase synthesis of amorphous Si/C nanoparticles as anode material for lithium ion battery. Journal of Alloys and Compounds 2021, 870, 159315.
  3. Kumar, P.; Berhaut, C. L.; Zapata Dominguez, D.; De Vito, E.; Tardif, S.; Pouget, S.; Lyonnard, S.; Jouneau, P. H., Nano-Architectured Composite Anode Enabling Long-Term Cycling Stability for High-Capacity Lithium-Ion Batteries. Small 2020, 16, (11), e1906812.
  4. Li, M.; Gu, J.; Feng, X.; He, H.; Zeng, C., Amorphous-silicon@silicon oxide/chromium/carbon as an anode for lithium-ion batteries with excellent cyclic stability. Electrochimica Acta 2015, 164, 163-170.

Reviewer 2 Report

The authors have resolved all my concerns and the manuscript can be published.

Author Response

Thanks Review 2 very much.